# Enrichment of Wholemeal Rye Bread with Plant Sterols: Rheological Analysis, Optimization of the Production, Nutritional Profile and Starch Digestibility

**DOI:** 10.3390/foods12010093

**Published:** 2022-12-24

**Authors:** Mussa Makran, Antonio Cilla, Claudia Monika Haros, Guadalupe Garcia-Llatas

**Affiliations:** 1Nutrition and Food Science Area, Faculty of Pharmacy, University of Valencia, Av. Vicente Andrés Estellés s/n, Burjassot, 46100 Valencia, Spain; 2Institute of Agrochemistry and Food Technology (IATA-CSIC), Av. Agustín Escardino, 7-Parque Científico, Paterna, 46980 Valencia, Spain

**Keywords:** Brabender farinograph, breadmaking, proximate composition, resistant starch, starch hydrolysis kinetics

## Abstract

Bread is one of the staple foods of many countries, and its enrichment with bioactive compounds is trending. This phenomenon is focused on breads with a good nutritional profile, such as wholemeal rye bread (WRB), in which enrichment with plant sterols (PSs) is allowed in accordance with European regulations. The objective of the present study was to optimize the production of a WRB enriched with PS (PS-WRB) and to evaluate the proximate composition and starch digestibility as an indicator of nutritional quality. The rheological analysis showed that the bread dough presents satisfactory farinographic properties (dough development time 6 min; stability 4 min; degree of softening 100 Brabender units) but high water absorption (67%). The PS-WRB is high in dietary fiber and low in protein (20.4 and 7.7% *w/w*, dry basis, respectively) compared with other cereals reported in the scientific literature. In turn, a low starch proportion was hydrolyzed during the simulated digestion (59.9% of total starch), being also slowly hydrolyzed, as deduced from the rapidly digestible starch value (56.5% of total starch). In conclusion, WRB is a suitable matrix for PS enrichment, which allows for obtaining a product with a good nutritional profile and potential health benefits.

## 1. Introduction

Noncommunicable diseases are responsible for 71% of deaths worldwide, the majority being cardiovascular, cancer, respiratory diseases and diabetes mellitus [1]. One of the most effective strategies to prevent these pathologies focuses on the dietary habits. In this sense, whole grain consumption has been associated with a lower risk of developing type 2 diabetes mellitus, cardiovascular diseases and some types of cancer. The beneficial effects linked to the intake of whole grain cereals are mainly attributed to their high content of dietary fiber [2]. In addition, the enrichment of cereal-based products with bioactive compounds is an increasingly used strategy to improve their functionality [3].

Among the different cereals, rye (*Secale cereale* L.) has the highest content of dietary fiber (18.7–22.2%) [4], and numerous bioactive compounds, such as phenolic acids, branched-chain amino acids and small peptides, have shown the physiological effects related to health [5]. The European Union has authorized the placing on the market of rye bread (≥50% rye flour and ≤30% wheat flour) enriched with plant sterols (PSs) [6,7]. PS are a group of bioactive compounds with a multitude of beneficial effects [8], such as the reduction (1.5–3 g PS/day) and maintenance (≥0.8 g PS/day) of blood cholesterol levels, accepted as health claims in the European Union [9,10]. Therefore, the enrichment of rye bread with PS implies an improvement in its functional properties. In this sense, a double-blind dietary intervention trial demonstrated that the intake of rye bread with added PS (2 g/day) for 2 weeks significantly reduced serum total and LDL-cholesterol (by 5.1 and 8.1%, respectively) compared with the control (rye bread without PS) [11].

For the development of a functional bakery product, it is important to evaluate the rheological properties of the flour to be used, the farinograph analysis being the most widely used for this purpose [12]. Once the characteristics of the bread dough have been determined, the breadmaking procedure must be optimized in order to obtain a product with adequate organoleptic characteristics. On the other hand, the evaluation of the proximate composition is fundamental to describe the nutritional value of foods. In addition, in starch-containing products, it is also of interest to assess the extent and kinetics of starch digestion as an indicator of the nutritional quality. There are different in vitro methods to evaluate starch digestibility, all based on monitoring starch hydrolysis during simulated gastrointestinal digestion [13,14,15]. The method by Englyst et al. [13] includes enzymes such as amyloglucosidase, invertase and pancreatin in the simulated digestion. Goñi et al. [14] use pepsin, in addition to glycolytic enzymes (amylase and amyloglucosidase). However, Guraya et al. [15] employ a simpler digestion, in which only amylase is used. The conditions (i.e., enzyme activity) contemplated in these digestion methods are not based on physiological data, and they have not used standardized parameters, thus reducing the reliability of the results obtained and preventing comparisons between studies. In this regard, a harmonized static simulated gastrointestinal digestion method has been developed by the COST Action INFOGEST network, in which an oral phase using salivary amylase is included for starch-containing foods [16]. Few studies have addressed the starch digestibility in several cereal-based foods using the INFOGEST method [17,18,19], none of them carried out on rye bread.

Recently, the use of human mastication has been proposed instead of the in vitro oral phase since it more resembles physiological conditions for the digestion of wholemeal rye bread (WRB) [20]. In addition, this study showed the higher bioaccessibility of PS in WRB, compared with that obtained in liquid matrices (17.1 vs. 8%).

Therefore, the objective of this study is to develop a PS-enriched WRB (PS-WRB) (1.8 g PS/portion) with optimal organoleptic characteristics and nutritional profile, in addition to the functional properties provided by its PS content. To achieve this goal, the farinographic analysis of the bread dough and the optimization of the baking process were carried out. Additionally, the characterization of the proximate composition and evaluation of the in vitro digestibility of the starch in the WRB during the application of the INFOGEST digestion was performed for the first time.

## 2. Materials and Methods

### 2.1. Materials

Commercial wholemeal rye flour was obtained from HARINERA LA META S.A. (part of La Meta Group, the Vall Companys Group’s flour division, Barcelona, Spain). An ingredient source of PS contained microencapsulated free PS (purity 74.7%, *w/w*) from tall oil (Lypophytol ME dispersible palm-free), and a blank ingredient without PS containing only the excipients used for the microencapsulation were provided by Lipofoods (Barcelona, Spain). L-ascorbic acid (purity ≥ 99.0%, *w/w*) and Celite were purchased from Merck LifeScience S.L.U. (St. Louis, MO, USA). Amyloglucosidase (*Aspergillus niger*) was obtained from Megazyme Inc. (Megazyme Ltd., Bray, Co. Wicklow, Ireland). Absolute ethanol (≥99.8%, *v/v*) was obtained from Panreac (Barcelona, Spain). Diethyl ether (≥99.5%, *v/v*), and potassium hydroxide and petroleum ether (boiling range 40–60 °C) were supplied by Scharlau (Barcelona, Spain). Water was purified (resistivity 18.2 MΩ × cm) using a Milli-Q system (Milford, MA, USA). For the assays of enzymatic activity and bile salt content and for the simulated gastrointestinal digestion, reagents were prepared according to Makran et al. [21].

### 2.2. Dough Rheological Analysis

The physical properties of the dough were analyzed using a farinograph (Brabender^®^ GmbH & Co. KG, Duisburg, Germany) according to the official method from the International Association for Cereal Chemistry (ICC) [22]. The thermostat was maintained at 30 °C, and 285.7 g of wholemeal rye flour (weight correction based on flour moisture) was placed in the farinograph bowl. Then, water was added to the flour until it reached a consistency of 500 Brabender units (Bus). The measurement was extended up to 12 min from the end of the development time. The following parameters were determined in the farinograph analysis: (i) water absorption—percentage of water required to yield a dough consistency of 500 BU; (ii) dough development time—the time (min) required for the midpoint of the curve to reach maximum consistency; (iii) stability—difference in minutes between the point at which the top of the curve first intercepts the 500 BU line and the point at which the top of the curve leaves the 500 BU line; and (iv) the degree of softening—the difference between the consistency (BU) of the center of the curve at the development time and the center of the curve at 12 min after the development time.

### 2.3. Breadmaking Procedure

The breads were prepared according to the formulation described by Sanz-Penella et al. [23], with modifications (reduction in sodium chloride content, from 2% to 1.6% flour basis, and the amount of flour, from 1000 to 300 g). The bread dough formula for PS-WRB consisted of wholemeal rye flour (300 g), compressed yeast (2.5% flour basis), sodium salt (1.6% flour basis), water (up to optimum absorption, 500 BU, 67% flour basis), ascorbic acid (0.01% flour basis) and the ingredient containing PS (4.3% flour basis). A PS nonenriched WRB was also made as a control, substituting the ingredient source of PS by the blank ingredient containing only the excipients (1.1% flour basis). Different conditions of the breadmaking procedure were tested to select the optimal method: ingredients were mixed for 5, 11 or 15 min (manually or with rotary blades), rested for 10 min, divided (four pieces of 100.7 or 102.5 g for WRB and PS-WRB, respectively) and hand balled and rested (15 min). Then, dough was proofed (45 min, 28 °C, 85% relative humidity), and the fermentation was monitored by measuring the increase in dough volume: 50 g pieces of the remaining doughs of WRB and PS-WRB were placed in graduated cylinders and left in the proofing chamber while the increase in dough volume was recorded periodically. Finally, the fermented doughs were baked (25, 30, 40 or 50 min, 170 or 180 °C).

### 2.4. Proximate Chemical Composition

The analytical measurements were carried out in triplicate. Analysis was performed with the partially dried milled bread, obtained after drying the bread in an oven at 24 °C overnight and subsequent grinding.

#### 2.4.1. Moisture

Determining the humidity of the bread products (>13% humidity) was carried out in two stages, according to the official method of the American Association of Cereal Chemists (AACC), with modifications [24]. The first step of drying is in a well-ventilated place at a controlled temperature, 24 °C (H_1_), whereas the second step of drying is at 130 °C during 1.5 h (H_2_). The total humidity of bread was calculated according to Equation (1):% Total moisture = [H_1_ + ((100 − H_1_) × H_2_)]/100(1)

#### 2.4.2. Ash

Ash content was measured according to official method [25]. Briefly, 3 g of sample was weighed in an incineration dish. Next, it was left to burn on a hot plate and then incinerated in a muffle furnace (Nabertherm GmbH, Lilienthal/Bremen, Germany) at 900 °C with air current for 3 h. Finally, the sample was left to cool in a desiccator for 1 h and weighed.

#### 2.4.3. Soluble and Insoluble Dietary Fiber

The soluble and insoluble dietary fiber contents were determined by a commercial kit K-TDFR-100A/K (Megazyme Ltd., Bray, Co., Wicklow, Ireland), according to the official method [26]. Briefly, 1 g of sample was weighed, and 40 mL of MES-TRIS buffer pH 8.2 was added. Next, 50 µL of thermostable α-amylase was added, and the mixture was incubated in a shaking water bath (SBS40, Cole-Parmer, Saint Neots, UK) (35 min, 100 °C). Then, 10 mL of water and 100 µL of protease were added, incubating in a water bath (30 min, 60 °C). Next, 5 mL of 0.561 M HCl was added, and the pH was adjusted to 4.1–4.8. The sample was incubated with 200 µL of amyloglucosidase in a water bath (30 min, 60 °C) and subsequently filtered in a crucible containing 1 g of Celite in kitasate flask (washing twice with 10 mL of water at 70 °C, 95% ethanol and acetone). The insoluble fiber was retained in the crucible, and the soluble fiber contained in the kitasate was precipitated with ethanol at 60 °C for 1 h and filtered in a crucible with Celite (washing twice with 15 mL of 78% and 95% ethanol and acetone, respectively). The crucibles were left to dry in an oven overnight at 103 °C and then weighed. Finally, the ash (incinerating during 5 h at 525 °C) and protein (Section 2.4.5) (Dumas method) contents were determined.

#### 2.4.4. Lipid

Lipid content was determined according to the official method of the AACC [27]. Briefly, 2 g of sample was weighed, and 2 mL of 95% ethanol and 10 mL of HCl (25 + 11) were added. After sample incubation (80 °C water bath, 90 min), 10 mL of 95% ethanol, 25 mL of diethyl ether and 25 mL of petroleum ether were added, stirred vigorously for 1 min. The sample was transferred to a separatory funnel, and the ethereal phase was transferred to a round bottom flask through filter paper (Whatman no. 2, 150 mm). Then, the solvent was evaporated under reduced pressure (800 mbar) and controlled temperature (40 °C), using a vacuum rotary evaporator (B490, BÜCHI Labortechnik, Flawil, Switzerland) (40 °C). Finally, the extract was dried in an oven (Binder, Tuttlingen, Germany) (100 °C, 90 min) and weighed.

#### 2.4.5. Protein

Protein determination was carried out by the Dumas combustion method, by applying a nitrogen conversion factor (5.83) according to the International Organization for Standardization/Technical Specification [28].

#### 2.4.6. Carbohydrate

The carbohydrate content was determined by difference according to the following Formula (2):Carbohydrate = 100 − [% (*w/w*) moisture + ash + total dietary fiber + lipids + proteins](2)

#### 2.4.7. Total Starch

The total starch content was determined by following the official method [29], using a commercial kit (K-TSTA 04/2009, Megazyme). Briefly, 100 mg of sample was weighed, and 0.2 mL of 80% ethanol and 2 mL of 2 M KOH aqueous solution were added, then incubated in a shaking ice/water bath (20 min). After adding 8 mL of sodium acetate buffer at pH 3, 100 µL of thermostable amylase and 100 µL of amyloglucosidase, the mixture was incubated (50 °C water bath, 30 min). Then, the sample was transferred to a 100 mL flask, and 1 mL was centrifuged (Sigma 3K15, St. Louis, MO, USA) at 1801× *g* for 10 min. Finally, 10 µL of supernatant were loaded into a 96-well plate with 290 µL of glucose oxidase-peroxidase (GOPOD) reagent and incubated (50 °C oven, 30 min). In parallel, a blank (10 µL of water + 290 µL of GOPOD) and a D-glucose control (10 µL of 1 mg/mL D-glucose + 290 µL of GOPOD) were prepared. Finally, the absorbance was measured at 510 nm in a multiwell spectrophotometer. The total starch content was not determined in the WRB because the starch digestion was tested on PS-WRB.

### 2.5. Simulated Gastrointestinal Digestion

#### 2.5.1. Determination of Enzymatic Activities and Bile Salt Content

The enzyme activity and bile salt content used for simulated gastrointestinal digestion was experimentally determined in a previous study [21], by following the guidelines provided in the INFOGEST method [16]. Regarding cholesterol esterase, the activity provided by the manufacturer was used because its use is not contemplated in the INFOGEST method, and there is no standardized methodology to determine its activity. Because the oral phase in the simulated digestion method was carried out by human mastication (see Section 2.5.2), the amylase activity in the saliva of the volunteer was determined. The collection of saliva was carried out following Sahu et al. [30]. The subject was instructed not to intake food or drink (except water) for at least 2 h prior to saliva collection; saliva accumulated in mouth cavity was collected and centrifuged (10,000× *g*, 10 min). After dilution of the supernatant (1/100 and 1/200, *v/v*) with sodium phosphate buffer at pH 6.9, the enzymatic activity was evaluated according to the 3,5-dinitrosalicylic acid assay proposed by the INFOGEST protocol [16].

#### 2.5.2. Digestion Procedure

To analyze starch digestibility, PS-WRB was subjected to simulated gastrointestinal digestion according to the INFOGEST 2.0 method adapted to PS-enriched foods [20]. To approximate the in vivo situation and factor in that the sample evaluated was a solid food, the in vitro oral phase was replaced by human chewing. Briefly, for the oral phase, 5 g of bread was chewed (40 chewing cycles, for ~36 s) by a healthy volunteer (41 years) with normal dentition, prior to informed consent. Then, the oral bolus (10 g) was subjected to the gastric phase, adding 7.5 mL of simulated gastric fluid, 0.98 mL of rabbit gastric extract (RGE) solution at 225 U/mL (final concentration of 60 U/mL gastric digesta, based on lipase activity) and 5 μL of 0.3 M calcium chloride, completing the pepsin activity provided by RGE to 2000 U/mL with 0.62 mL of pepsin solution at 25,000 U/mL. After mixing manually for one minute, the pH was adjusted to 3, and water was added up to a volume of 20 mL. Then, the gastric mixture was incubated in a shaker bath (2 h, 37 °C and 95 rpm). To carry out the intestinal phase, 11 mL simulated intestinal fluid, 5 mL pancreatin solution at 800 U/mL (final concentration of 100 U/mL intestinal digesta, based on trypsin activity), 40 μL of 0.3 M calcium chloride, 2.5 mL bile solution at 166 mM bile salts (final concentration of 10 mM in the intestinal digesta) and 0.1 mL of the 30 U/mL CE solution (final concentration of 0.075 U/mL in intestinal digesta) were added. The intestinal mixture was stirred manually for 1 min, adjusted to pH 7, and ultrapure water was added to a final volume of 40 mL. Finally, the mixture was incubated in a shaking bath (2 h, 37 °C and 95 rpm). The blanks of digestion (5 g water) were carried out in order to subtract the glucose content in the digesta from the digestion reagents. Digestions were carried out in triplicate.

### 2.6. Kinetics of Starch Hydrolysis

In order to monitor starch digestion, glucose was quantified (GOPOD FORMAT, K-GLUC 09/14, Megazyme) in the undigested sample (PS-WRB), oral bolus and intestinal digesta, at different times (0, 10, 20, 40, 60, 80, 100 and 120 min), by following Azizi et al. [31], with some adaptations. Briefly, 1 or 2 g of bread or oral bolus was weighed, and 6 or 7 mL water was added, respectively. After sample centrifugation (3600× *g*, 30 min), 100 µL of supernatant or intestinal digesta was incubated with 400 µL ethanol for 30 min, with agitation by using a sample tube rocker (KS 260 Basic, Ika, Germany). Then, the sample was centrifuged (4000× *g*, 15 min), and 100 µL of supernatants was incubated with 500 µL of amyloglucosidase solution (27 U/mL, in a sodium acetate buffer, pH 4.8) at 37 °C for 1 h. Then, 10 µL of sample was loaded into a 96-well plate with 290 µL of GOPOD reagent and incubated for 30 min at 50 °C. Finally, the absorbance was measured at 510 nm. The rate of starch digestion was expressed as the percentage of total starch present in sample hydrolyzed at different times. Starch digestion in the intestinal phase was described using the following nonlinear model: C_t_ = C_0_ + C_∞_ (1 − e^−kt^), where C_t_ is the percentage of hydrolyzed starch at time t, C_0_ is the percentage of starch hydrolyzed in the oro-gastric phase, C_∞_ is the percentage of hydrolyzed starch in the intestinal phase and k is the constant (min^−1^). Starch fractions based on their digestibility were determined (as represented graphically in Figure 1): rapidly digestible starch (RDS) (digested from beginning to the 20 min of intestinal phase), slowly digestible starch (SDS) (digested between 20 and 120 min of intestinal phase) and resistant starch (RS) (undigested in the digestion) [17].

### 2.7. Statistical Analysis

A Student *t*-test (α = 0.05) was used to analyze differences in the proximate composition between PS-WRB and WRB and starch hydrolysis in the oral phase between independent assays. Statgraphics Plus 5.1 software (Statpoint Technologies Inc., Warrenton, VA, USA) was used for statistical analysis.

## 3. Results and Discussion

### 3.1. Farinographic Properties

The development of a cereal-based functional food contemplates as a first step the analysis of its raw material. The farinographic analysis is useful to evaluate the rheological characteristics of the bread dough [12], as a predictor of bread quality [32]. In this sense, the farinographic analysis of the bread dough obtained from the wholemeal rye flour showed the following features: water absorption—67%; dough development time—6 min; stability—4 min; degree of softening—100 BU.

Water absorption is one of the most important characteristics of bread dough and is closely related to the quality of bakery products. In the present study, the wholemeal rye flour exhibited high water absorption, within the observed range for wholemeal rye flours (54.0–66.8%) [33,34]. The high content of dietary fiber could explain the elevated water absorption of the flour, even more so given that the characteristic dietary fiber profile of rye (mainly arabinoxylan) provides a high water-retention capacity [35]. On the other hand, previous studies have demonstrated that dough made of rye flour exhibits very low resistance to mechanical treatment, as indicated by a very short dough development time (0.6 min), its stability (0.5 min) and its significant softening (175 BU) [34]. However, the wholemeal rye flour showed more-satisfactory farinographic parameters in the present study. It has been reported that the different milling technologies can generate differences in the farinographic parameters of wholemeal rye flours [33], which would explain the discrepancy observed in the mechanical resistance parameters. In relation to the dough development time, a high value was observed in comparison with the wheat flour reported by other authors (2.5 min) [36]. In the same study, the supplementation of wheat flour with rye bran increased the dough development time by 6.5 min, which was attributed to the coarseness of the rye bran. On the other hand, the stability is another parameter related to flour strength. Dough is considered to have good resistance when the stability time ranges between 4 and 12 min 6 min being the ideal [37].

In summary, the data indicated that the farinographic properties of the bread dough obtained with the wholemeal rye flour is suitable and optimal for the development of a functional rye-based bread.

### 3.2. Optimization of the Breadmaking Procedure

Different factors of the breadmaking process significantly affect the features of the bakery products, from kneading to fermentation and baking [38]. Thus, determining the optimal conditions of the baking procedure is key to guaranteeing a product with adequate characteristics and that is acceptable to the consumer. To select the optimal breadmaking process, different conditions were tested. As criteria for selecting the optimal method, the temperature of the bread dough after kneading (optimal 24 °C), the baking extent of the bread and its organoleptic characteristics (color and texture) were considered. Kneading with rotating blades for 11 min was selected because it allowed for obtaining the optimal temperature of the dough and a homogeneous dough. Finally, the baking conditions that provided a product with good baking extent and with apparently good organoleptic characteristics were selected (25 min, 180 °C). The lowest temperature resulted in an undercooked dough, and longer times resulted in an excessively tough crust. From each bakery, four breads with an average weight of 81.3 ± 1.1 g were obtained. Figure 2 shows photographs of the bread from different perspectives.

### 3.3. Proximate Chemical Composition

In order to determine the nutritional quality of raw material and breads, the chemical compositions of the flour, WRB and PS-WRB were determined (shown in Table 1). The proximate composition of wholemeal rye flour agreed with the data reported by other authors (% *w/w*, dry basis: 58% starch, 12% proteins, 2% lipids, 2% ash and 18% total dietary fiber) [39]. The largest difference was observed in protein content, which is attributed to a different nitrogen conversion factor. In addition, the proximate composition between WRB and PS-WRB were not statistically significant (*p* > 0.05), except in moisture and lipid contents, thanks to PS enrichment.

Beyond the quantitative value of the nutrients in the flour, it is worth addressing its quality compared with other cereals. This approach will allow us to understand the rest of the results obtained in the study, from the rheological behavior of the dough to the kinetics of starch digestion.

#### 3.3.1. Protein

Evaluating the protein content is of interest because it provides information on the nutritional value and is closely related to the rheological properties of the dough. The data obtained show that both wholemeal rye flour and its bread have low protein content (7.7 and 7.8–7.9%, respectively). In this sense, rye and wholemeal rye bread showed in previous studies lower protein content than did other breads (multicereal, wholemeal wheat and oat) (5.7–6.6% vs. 7.3–9.5%, fresh weight) [40]. In addition, different studies have demonstrated that whole rye grain presents lower protein content than that of other whole cereals, such as wheat, barley, oat or buckwheat (11.1% vs. 13.9–17.8% *w/w*, dry basis) [41]. On the other hand, a high level of lysine (the main limiting amino acid in cereals) in rye grain, in comparison to wheat and triticale (3.5, 3.2 and 3.0 g/kg, dry basis, respectively) [42], has been reported. This could suggest that despite the lower protein content in rye, it presents a better amino acid profile. In addition to their low content, the proteins of the rye flour do not form an optimal structure during the kneading of the dough [43]. The high arabinoxylan content contributes to delaying the formation of the gluten network, reducing the elasticity and stability of the dough. This impairs gas retention, which would explain the low volume increase of the dough during fermentation observed in the present study. Although rye proteins do not play an important role in the formation of the dough structure, they are important for the typical flavor of rye bread [44].

#### 3.3.2. Lipid

Lipids provide a variety of beneficial properties during bread processing and storage, in addition to driving nutrients and lipophilic bioactive compounds [45]. One study reported lower lipid content in whole grain rye than in other whole cereals (wheat, barley, oat and buckwheat) (1% vs. 1.2–3.1% *w/w*, dry basis) [41]. In addition to the lower lipid content, a good nutritional profile of the lipid fraction of rye has been reported, observing a higher amount of total unsaturated fatty acid (81.5% vs. 77.9–80.1%, percentage relative to the total fatty acids content) in comparison to other cereals (wheat, triticale, barley and oat). The linoleic, oleic and linolenic acids (55.0%, 19.4% and 6.7% of total fatty acids, respectively) [46] stand out as major fatty acids. This implies that rye not only has low fat content but also has a more beneficial fatty acid profile than other cereals. In this sense, according to the authorized nutritional claims in the annex of Regulation (EC) No. 1924/2006, rye flour and WRB could be declared as low-fat-content food (<3 g fat per 100 g) [47]. However, in the PS-WRB, enrichment with PS increased the lipid content, exceeding the threshold for such a claim. Moreover, the high value of the unsaponifiable fraction of rye lipids should be highlighted. In relation to fat-soluble vitamins, Zielinski et al. [41] determined the content of vitamin E in different cereal grains, revealing the higher content in whole grain rye (22.0 vs. 4.7–13.3 IU/kg). Other important components in rye are the PS. Higher levels of total PS have been observed in rye than in wheat, both in the whole grain (99 vs. 78 mg/100 g, dry basis) [48] and in bread (119–136 vs. 61 mg/100 g, dry basis) [49]. With these data, it can be concluded that rye not only has a nutritionally adequate fatty acid profile but also has a significant amount of bioactive compounds and lipophilic vitamins.

#### 3.3.3. Carbohydrate

Carbohydrates are the most abundant macronutrients in rye, the majority being starch. Starch content was reported to be lower in rye flour than in wheat (73 vs. 80% *w/w*, dry basis), with a comparable proportion of the amylose (relative to amylopectin) (27–29%) [50]. In addition to quantitative differences, structural differences have also been observed in the same study. Specifically, rye presented a lower proportion of type-B granules (≤9.3 μm in diameter) (10%–15% vs. 20%) and a larger average particle size for type-A (up to 62.5 μm in diameter) (31 vs. 26 μm). These structural differences could justify in part the differences in the kinetics of starch hydrolysis between both cereals (addressed in Section 3.4). On the other hand, it has been observed that starch does not represent all the carbohydrate content in rye flour and bread (see Table 1), indicating the presence of other nonstarch carbohydrates. In this regard, the presence of monosaccharides (arabinose, xylose and galactose) at 5.1% (*w/w*, dry basis) in rye flour has been reported [50].

A quantitative analysis of carbohydrate, and especially starch content, provides useful information on nutrition. However, beyond the starch content, its digestibility can become a better nutritional indicator because it is related to the glycemic response [51].

#### 3.3.4. Dietary Fiber

Dietary fiber is one of the most characteristic components of rye, responsible for the technological [52] and functional [53] properties of rye-based products. In the present study high dietary fiber content was observed in the wholemeal rye flour and breads, reaching amounts that justify the nutritional claim that it is high in fiber (>6 g of fiber per 100 g), according to European regulations. In addition, insoluble dietary fiber in wholemeal rye flour represented 71% of the total dietary fiber content, values similar to those observed by other authors (83%) [39]. Regarding the fiber profile, the main types of dietary fiber in the rye are arabinoxylans, β-glucans, cellulose, lignin and fructans [54]. In this sense, the differences in the dietary fiber content between the wholemeal rye flour and wheat flour of a similar degree of extraction are attributed to the higher levels of fructans, β-glucans and arabinoxylans, while cellulose and lignin are found at comparable levels.

As mentioned before, dietary fiber plays a great role in the health properties of rye. It has been suggested that one of the main beneficial effects of rye, the prebiotic action, is related to the content of arabinoxylans [55]. In summary, rye is characterized by high dietary fiber content, which provides numerous health benefits. However, dietary fiber negatively affects the technological properties of rye, reducing the quality of bread dough.

### 3.4. Starch Hydrolysis Kinetics

The analysis of the proximate composition allows us to evaluate the nutritional value of a food and its raw materials. This aspect is of interest in a functional food because in addition to the biological effects derived from its content in bioactive compounds, it must present an adequate nutritional profile. Despite all of this, the nutrient content in isolation does not provide complete information on the nutritional quality of a food, because other aspects, such as digestibility, are decisive. In this sense, in cereal-based functional foods, such as PS-WRB, it is of interest to evaluate the hydrolysis kinetics of starch and its different fractions as indicators of nutritional quality. In this regard, starch is classified as RDS, SDS and RS on the basis of its digestibility extent, which depends largely on the structure of the starch and its processing. RDS is rapidly digested and absorbed in the proximal regions of the small intestine, causing a rapid rise in blood glucose. Blood glucose fluctuations can induce stress on glycemic homeostasis systems, with negative health consequences [51]. On the contrary, SDS is digested slowly throughout the small intestine to provide a sustained release of glucose. Due to its association with a stable glucose metabolism, different potential health benefits have been attributed to this starch fraction [56]. On the other hand, RS is not digested in the upper gastrointestinal tract, so it reaches the colon, where it can be fermented by the microbiota. With this, short-chain fatty acids can be produced that provide beneficial effects for health [57].

The results of starch hydrolysis in the different phases of simulated gastrointestinal digestion are presented in Figure 3, which shows the data obtained for starch hydrolyzed at intestinal phase during in vitro digestion, with the following adjusted equation for hydrolysis kinetics: C_t_ = 31.114 + 28.8317 (1 − e^−0.1064t^) (R^2^ = 0.9881).

#### 3.4.1. Oral Phase

Starch digestion begins in the mouth with the action of salivary amylase [58], an enzyme capable of continuing to act in the gastric stage, although with less activity because of the pH of the gastric environment [59]. Because of the heavy involvement of salivary amylase in starch digestion, the activity of amylase in the saliva of the volunteer for the oral phase was evaluated. The results show an activity of 266.3 ± 2.5 U/mL saliva, which means that the activity in the oral bolus would be 133 U/mL, after an optimal food–saliva mixture (1:1, *w/w*). This value is higher than that reported by other authors in subjects of the same age as the volunteer (40–60 years; 76–159 U/mL saliva) [30]. It is also higher than the value recommended by the INFOGEST method for the simulated oral phase (75 U/mL of oral bolus) [16]. Numerous factors that affect amylase activity in saliva could explain this discrepancy, including the stress to which the volunteer may have been subjected, known to increase amylase activity in saliva [60].

Regarding the hydrolysis of starch in the oral phase, two independent assays were carried out, in which a different food–saliva ratio was obtained. In one of them, the recommended ratio of 1:1 (*w/w*) was reached, which indicates that 50% of the weight of the oral bolus corresponded to saliva. In the second, the lower degree of salivation of the subject resulted in an oral bolus with less saliva (36%, *w/w*). The lower presence of saliva was related, as expected, to less starch hydrolysis (39 ± 3 vs. 23 ± 3%, *p* < 0.05). These data indicate that the degree of salivation show an intraindividual variability, which has an impact on the oral hydrolysis of starch.

Compared with the results from other authors, the hydrolysis extent was highly superior to those observed in other studies that apply the in vitro oral phase according to the INFOGEST method (75 U amylase/mL): white bread (5.1%), gluten-free white bread (2.4%) [17], baked bread (8–9%), steamed bread (8–10%) and French baguettes (8–10%) [17]. In addition, our data were higher than 13% hydrolyzed starch during in vivo chewing of white bread [61]. The greater hydrolysis reported in our study could be due to the greater amylase activity.

#### 3.4.2. Gastric Phase

The oral bolus obtained after human mastication was subjected to a simulated gastric phase. The oral bolus with 36% saliva was used since it presents amylase activity closer to that recommended by the INFOGEST method. After the gastric phase, the data reported a 31% ± 5% of digested starch (starch hydrolysis in the end of gastric phase can be also obtained from the kinetics model equation C_0_ = 31.114). This indicates that part of the salivary amylase activity is preserved during the gastric phase. According to the data, salivary amylase acts in the oral phase by hydrolyzing 23% of the starch in just 2 min. When the oral bolus passes into the gastric phase, there is a sudden drop in pH, reducing the activity of salivary amylase, which under these suboptimal conditions acts by hydrolyzing 8% of the starch during the 2 h. Contrary to what was seen in the oral phase, hydrolysis in the gastric phase was lower than that reported in previous studies on white bread (44%), gluten-free bread (23%) [17], baked bread (50–55%), steamed bread (35–45%) and French baguettes (45–60%) [18]. One of the reasons that could explain the lower starch digestion in the gastric phase is the particle size, which is known to be smaller in whole rye bread than in wheat bread [62]. The small particles are associated with less starch hydrolysis in the gastric phase because the easy diffusion of gastric fluid could rapidly inactivate of salivary amylase [18], which would justify the lower gastric starch hydrolysis in PS-WRB.

#### 3.4.3. Intestinal Phase

In the intestinal phase, starch hydrolysis continued until the 30 min stage was reached. During this time, 28.8% (C_∞_) of the starch was hydrolyzed, a value much lower than that observed for white bread (37.9%) and gluten-free bread (51.1%) [17]. This again indicates that the starch contained in PS-WRB had a lower digestibility. On the other hand, total starch hydrolysis was calculated by adding the hydrolyzed starch during the gastric oral phase (C_0_) and the adjusted equilibrium percentage of hydrolysis during the intestinal step (C_∞_). The obtained value (59.9%) was again lower than that reported for white bread (87%), gluten-free bread (76.5%) [17] and French baguettes (95%) [19]. Therefore, the remaining starch corresponds to RS (40.1%). On the other hand, the starch hydrolysis kinetics was determined by factoring in the hydrolysis in the intestinal stage in that the rate of glucose absorption in the small intestine determines the glycemic response to starchy foods. In this sense, a major part of the digestible starch was hydrolyzed at 20 min of the intestinal phase (56.5%) (corresponding to RDS), whereas the remaining 3.4% was digested in the following 10 min (SDS). The proportion of RDS in PS-WRB was similar to that reported by Ribes et al. [18] for French baguettes (55%) but lower than wheat bread (79%) and gluten-free bread (74%) [17]. This suggests that the starch in PS-WRB is not only less digested but also digested much more slowly.

These results agree with in vivo studies showing a lower postprandial insulin response after the intake of rye bread compared with wheat bread [63,64], attributed to a slower intestinal uptake of glucose [63]. This is confirmed in the present in vitro study, showing a lower and slower glucose release during digestion of the starch contained in PS-WRB than those of other breads reported in the scientific literature.

Many possible reasons could explain the low digestibility of the starch in the PS-WRB. The first could be related to the structure of bread. Juntunen et al. [63] reported that rye bread presents a more compact structure than that of wheat bread, which could decrease starch accessibility for amylases, explaining its low digestibility. On the other hand, the lowering effect on starch digestion by dietary fiber is known [65]. In this sense, the content of dietary fiber in WRB is higher than that in wheat bread [54]. In addition, it has been suggested that the inhibitory effect on amylase by phenolic compounds of rye could be responsible for the lower insulin response to the intake of rye bread [64].

## 4. Conclusions

In the development of a cereal-based functional food, it is essential to verify the suitability of the raw materials used. In this sense, the farinographic analysis of the whole rye flour shows acceptable properties, but that could be improved in future studies by incorporating ingredients that provide elasticity to the dough. On the other hand, using a wholemeal flour allows for making the most of its nutritional properties, which in this case are its high fiber content, good fatty acid profile and high protein quality. Moreover, enrichment with PS enhances the functional properties of the final product. Finally, in addition to the optimal nutritional composition of rye bread, a good level of digestibility is observed in starch. In this sense, the low degree of digestibility, which provides a great source of resistant starch, to which numerous beneficial effects for health are attributed, must be highlighted. In this way, the slow release of glucose during digestion could lead to a low postprandial insulin response. In short, WRB is a suitable matrix for enrichment with PS thanks to its properties. However, a tasting study is necessary to evaluate the organoleptic characteristics of the product and assess whether an improvement in the formulation is needed in order to guarantee the acceptability of the bread to consumers. Other aspects remain to be evaluated, such as the effect of PS on starch digestibility. The evaluation of starch hydrolysis in nonenriched bread (to compare it with enriched bread) would provide information on this and on the possible effect of these bioactive compounds on starch digestibility and possibly on the postprandial glycemic and/or insulinemic response.

## Figures and Tables

**Figure 1 foods-12-00093-f001:**
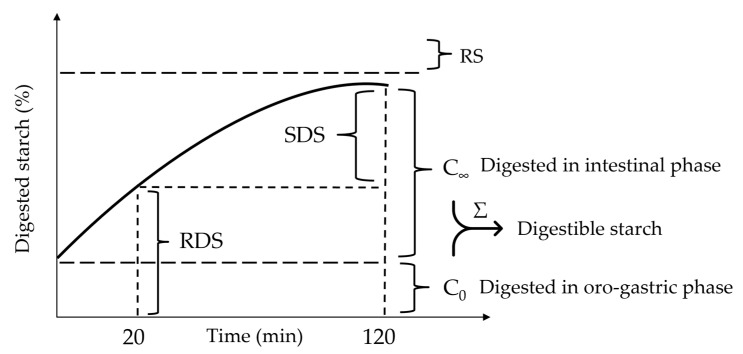
Graphical representation of in vitro digestion of starch during intestinal phase. RDS—rapidly digestible starch; SDS—slowly digestible starch; RS—resistant starch.

**Figure 2 foods-12-00093-f002:**
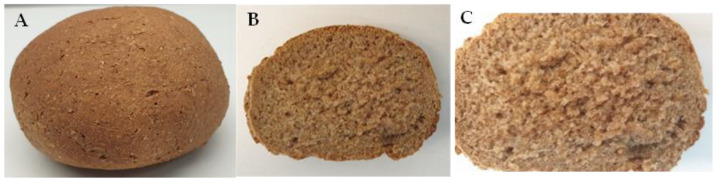
Photographs of wholemeal rye bread: (**A**) whole bread, (**B**) central slice and (**C**) crumb.

**Figure 3 foods-12-00093-f003:**
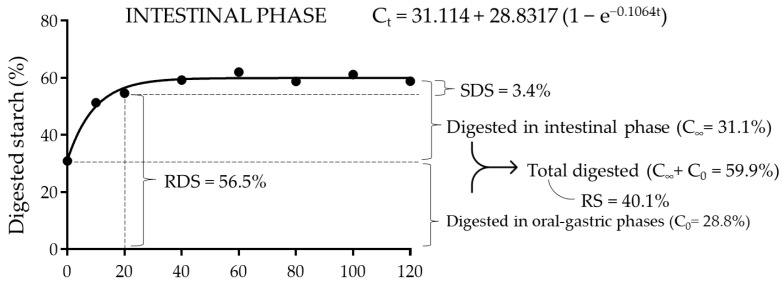
Adjusted curve of starch hydrolysis during the in vitro intestinal phase according to following the following nonlinear model: C_t_ = C_0_ + C_∞_ (1 − e^−kt^). C_t_ is the percentage of hydrolyzed starch at time t; C_0_ is the percentage of starch hydrolyzed in the oral-gastric phase; C_∞_ is the percentage of hydrolyzed starch in the intestinal phase; and k is the constant (min^−1^). The total digested starch was calculated by adding the hydrolyzed starch during the gastric oral phase (C_0_) and the adjusted equilibrium percentage of hydrolysis during the intestinal step (C_∞_). Resistant starch corresponds to starch not digested during in vitro digestion. Rapidly digestible starch (RDS) corresponds to starch digested from the beginning to the end of the 20 min intestinal phase. Slowly digestible starch (SDS) corresponds to starch digested between 20 min and 120 min of the intestinal phase. Resistant starch (RS) corresponds to undigested starch.

**Table 1 foods-12-00093-t001:** Proximate composition (% *w/w*, dry basis) of flour: WRB and PS-WRB.

	Flour	WRB	PS-WRB
Moisture	11.5 ± 0.3	26.14 ± 0.02	32.7 ± 1.0 *
Protein	7.7 ± 0.5	7.9 ± 0.1	7.8 ± 0.1
Ash	1.54 ± 0.01	1.99 ± 0.03	2.0 ± 0.1
Lipid	1.6 ± 0.1	1.57 ± 0.03	4.7 ± 0.2 *
Carbohydrate	73.0 ± 0.5	69.7 ± 1.8	65.2 ± 0.6
Total starch	59.1 ± 2.4	–	54.7 ± 2.7
Insoluble fiber	11.5 ± 0.2	14.5 ± 1.8	15.2 ± 1.8
Soluble fiber	4.7 ± 0.5	5.4 ± 0.2	5.1 ± 0.2

Data are shown as mean values ± standard deviation (*n* = 3). WRB—wholemeal rye bread. PS-WRB—plant sterol–enriched wholemeal rye bread. The asterisk (*) indicates statistically significant differences (*p* < 0.05) between WRB and PS-WRB.

## Data Availability

Data presented in this study are available on request from the corresponding author.

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
