# Peer review of "Enrichment of Wholemeal Rye Bread with Plant Sterols: Rheological Analysis, Optimization of the Production, Nutritional Profile and Starch Digestibility"

_foods, 2022, doi:10.3390/foods12010093_

Round 1
Reviewer 1 Report
Dear Authors,
I provide some comments to the reviewed typescript.
Kind regards
Reviewer
Keywords: Brabender farinograph; bread-making; cereal; proximate composition; resistant starch; starch hydrolysis kinetics
Introduction: Please provide information about the relationship between the enrichment in plant sterols and the improvement of the nutritional value of the final product. Does subjecting the additive to fermentation by yeast, high temperature in the baking chamber affect the content of PS in the final product (bread). Does the role of this additive affect starch digestibility? Or the availability of PS in the finished product and the enrichment of the diet in PS for a potential consumer of PS-WRB bread?
The aim of the research is not sufficiently explained.
Line 38: Please add information about the average content of fiber in rye grain.
Line 44: What dose or intake of PS causes in the listed results?
Line 102: Were the farinographic properties of the dough made of flour with the participation of PS tested?
Line 104-106: The authors used different conditions for the preparation and fermentation of rye dough: the method of kneading, dough maturation and baking. Such a multi-faceted approach is valuable and will allow optimizing the method of dough preparation and baking also in terms of large-scale production. However, in the case of a rye dough, and even more so a wholemeal rye flour dough, it is extremely advantageous to use a phase guiding of the dough, e.g. using a sourdough phase. I suggest that the authors also consider such issues in the future. Then it is possible to add PS only in the last phase, i.e. dough preparation.
The authors reported that: “Fermentation was monitored by measuring the increase in dough volume: 50 g pieces of dough were placed in graduated cylinders and left in the proofing chamber while the increase in dough volume was recorded periodically.” Does this description apply to dough fermentation: Then, dough was proofed (45 min, 28ºC, 85% relative humidity)…
Line 118: How is the end of fermentation determined?
Line 277: In summary, the data indicated that the viscoelastic fariographic properties of the bread dough
Lines 287 and 289: cooking? baking
Line 280-293: Optimization of the bread-making procedure
Indicating the optimal values among several variants taken into account is too laconic. Why were the kneading time, temperature and baking time chosen?
Line 329: This paragraph discusses the lipid content of PS enriched bread? In connection with the enrichment of bread in PS, the question arises: did and how did the PS content change during dough fermentation and baking?
Subchapter 3.4: The digestibility of starch in rye bread and in rye bread enriched with PS is well discussed.
Author Response
The reviewer response document is attached.

Reviewer 2 Report
This manuscript was entitled “Enrichment of wholemeal rye bread with plant sterols: Rheological analysis …”. It is an interesting study; however, the objective of applying plant sterols were not explained in this manuscript. Whether there were similar research and the gap in previous research were not introduced either. The novelty of the present study also needs to be more clearly stated in the introduction as well.
Other comments:
1. Abstract, “a low and slow in vitro digestion of starch was observed, as deduced from the percentage of digestible starch (59.9%) and rapidly digestible starch (56.5%).”
Please clarify what “a low in vitro digestion of starch” refer to. And how to define digestible starch? Total digestible starch?
2. Section 3.3.1, the protein content of the three samples needs to be described and discussed in the main text.
3. Section 3.3.3, please clarify what the “carbohydrate” refers to, as starch and dietary fiber are also carbohydrates.
4. Key results need to be summarized in the results.
Author Response
The reviewer response document is attached
